# ‘24/7’ Caregiving: A Qualitative Analysis of an Emerging Phenomenon of Interest in Caregiving

**DOI:** 10.3390/ijerph192417046

**Published:** 2022-12-19

**Authors:** Esther Yin Hui Chew, Zhi Lei Ong, George Frederick Glass, Ee-Yuee Chan

**Affiliations:** 1Nursing Research Unit, Nursing Service, Centre for Healthcare Innovation, Tan Tock Seng Hospital, Singapore 308443, Singapore; 2Geriatric Education and Research Institute (GERI), Singapore 768024, Singapore

**Keywords:** family caregiver, caregiving, dementia, older persons, caregiving burden, COVID-19, stress

## Abstract

The COVID-19 pandemic has exacerbated the difficulties faced by caregivers who have to provide continuous ‘24/7’ care to persons with dementia with minimal formal and informal support. While caregivers have reported heightened levels of caregiving distress and burden during the pandemic, there remains a dearth of research pertaining to their lived experiences of providing continuous care with little respite and the corresponding physical, psychosocial and emotional impacts of caregiving ‘24/7’. The present study uses data obtained from interviews with dementia caregivers (N = seven) that were collected as part of a larger study on Carer Matters, a hospital-based holistic caregiver support program held during COVID-19, to conduct a secondary thematic analysis. The findings revealed three themes that defined the shared experiences of ‘24/7’ caregivers: (1) A World Overturned, which refers to the increase in caregiving intensity and burden due to the pandemic; (2) Burning on Both Ends, which refers to the impossible balance between caregiving and their personal lives; and (3) At Wits’ End, which refers to an overwhelming sense of hopelessness and helplessness over their caregiving situation. These findings highlight the challenges and unsustainability of ‘24/7’ caregiving and the detrimental impact that round-the-clock care wields on caregivers’ physical and mental well-being. Implications and recommendations are discussed in accordance with the cultural particularities of the study’s Asian context (Singapore), with calls for greater caregiver support to be better integrated into society and the community, especially at the neighborhood and grassroots level, to alleviate caregiving burden and safeguard their well-being.

## 1. Introduction

The proportion of older persons (above 65 years old) in Singapore is disproportionately increasing and is projected to grow to one in four persons by 2030 [1]. More crucially, 10% of people aged above 60 years in Singapore are reported to have dementia, and dementia cases are expected to surge with the oncoming aging population as well, despite the reduced number of family members providing care for this population [2,3]. Indeed, families are primarily the cornerstone of support for persons with dementia (PwD) as they play an essential role in maintaining their health and well-being. However, many caregivers often struggle with the responsibility [4,5] as they have to manage both behavioral symptoms and emotional and physical care needs, ‘tasks’ that are often repetitive, strenuous, ‘dirty’ and time consuming [6].

A subset of this group often finds themselves expected to monitor their care recipients closely and labor on caregiving tasks perpetually, leading to a life revolving around the caregiving role ‘24/7’ [6]. This resembles a relentless and all-consuming affair that has physical and psychosocial ramifications on family caregivers caring for their loved ones, wielding significant negative impacts on their mental and physical health and overall quality of life [7,8]. The onset of caregiver stress is often insidious, with the caregiver incrementally absorbing more roles and responsibilities to assist their older loved one until they are fully subsumed into the caregiving role [9]. Indeed, many ‘24/7’ caregivers often find themselves unable to mentally ‘switch off’ from caregiving as they worry about their older care recipient even when they are not ‘doing’ caregiving and, consequently, they become physically, emotionally and mentally engulfed by their caregiving duties and responsibilities ‘24/7’. This points to the multifaceted nature of our definition of the ‘24/7’ caregiver, which is a multidimensional and complex construct, encompassing not only caregiving and its physicality but also its emotional and psychological domain [5]. This is prevalent in Singapore, where nearly 95% of family caregivers live together with their care recipients in the same household, where care provision is a cultural norm and filial obligation is the predominant cultural value in relation to the caregivers’ caregiving experiences [10,11]. However, such caregiving is concerning, because if it is unrestrained and unaddressed, it could spiral out of control, especially if the caregiver has no recourse to any form of relief. This would lead to caregiver collapse under the pressure of their inability to sustain caregiving for their loved one, which would unfortunately and inevitably lead to the institutionalization of their older loved one [12].

Such caregiving experiences have been intensified over the last two years due to the COVID-19 pandemic. The tightened restrictions imposed by governments to contain the spread of COVID-19 has intensified caregiving stresses through lockdowns and physical distancing measures, which contributed to challenges regarding social support for caregivers of PwD [13]. The implementation of stay-at-home orders and the suspension of community-based day-care services and social gatherings reduced the amount of support and respite care that caregivers usually receive that often help alleviate caregiver stress [14], which thereby increased the intensity of their caregiving responsibilities and frustrations as they had to provide full-time care without their usual respite, which was especially apparent during the pandemic period.

Despite growing recognition that caregiving demands often parallel a ‘24/7’ job [15], there is still little empirical research examining the relationship between intensive and prolonged caregiving (‘24/7’ caregiving) and the dementia caregiver’s physical and psychosocial well-being during COVID-19 in an Asian population (Singapore).

## 2. Materials and Methods

A secondary analysis was conducted on data collected from caregivers of PwD who were involved in a holistic caregiver support program during the COVID-19 pandemic [16,17]. Data were collected in a tertiary hospital in Singapore between December 2020 and November 2021.

The inclusion criteria were as follows: main family caregiver aged 21 and above and providing home-based care. Participants whose care recipients were planned for discharge to a long-term care facility were excluded from the study.

Ethics approval for our primary study was obtained from the National Healthcare Group Domain Specific Review Board (Ref: 2020/00087). According to a set of criteria based on the researcher’s expertise, similarities in the research purpose, methodology utilized and type of information collected, we evaluated the primary dataset to ensure its appropriateness and congruence for the new research question [18]. In the primary study, eligible family caregivers of PwD admitted into the wards piloting the caregiver support program were referred to the study team by the attending ward clinical staff. Subsequently, the study team approached the caregivers to explain the purpose of the study and addressed any questions before obtaining written informed consent. In-depth semi-structured interviews were conducted to elucidate the participants’ lived caregiving experiences and perspectives in an insightful and detailed manner while affording the interviewers flexibility in probing or clarifying and exploring other subjects of interests [19]. In lieu of the COVID-19 pandemic, its accompanying restrictions and confidentiality purposes, the interviews with the participants were conducted online via Zoom in the privacy of their own homes. The interviews were recorded, transcribed verbatim and reviewed for accuracy. Each interview took approximately 20 to 60 min to complete.

E.Y.H.C. and Z.L.O. used thematic analysis to explore the caregivers’ experiences of caring for their loved ones during COVID-19 and identified salient common characteristics of ‘24/7’ caregivers in the transcripts. Although E.Y.H.C. and Z.L.O. did not participate in the primary data collection, frequent communications were established with the primary study team (G.F.G.J. and E.Y.C.) to discuss the primary study, their participant observations and the interviews’ context, thereby facilitating sensitivity to the context of the primary study and bridging the gap between them through deeper immersion [20]. This involved several stages of coding and reduction. E.Y.H.C. and Z.L.O. read the transcripts repeatedly and thoroughly to ensure deep immersion and familiarization with the primary study data. Co-coding took place, with initial codes generated inductively by E.Y.H.C. and Z.L.O. independently until sufficient data were gathered to derive important conclusions and further coding produced no further valuable insights. All relevant text units relating to the research topic were subsequently identified, extracted and assigned descriptive codes. The codes were then collated and condensed into themes at higher levels of abstraction, enabling broader themes to emerge from the data [21]. Broad overarching themes were identified and defined and were constantly reviewed by the team during regular meetings. Established aspects of trustworthiness contributed to the study’s rigor [22]. Credibility was enhanced through the usage of field notes and member checking. An audit trail was maintained throughout the discussion process, chronicling down all analytic and methodological decisions.

## 3. Results

The participants (N = seven) in this study were (1) retired or working part time, (2) had no additional support (e.g., domestic helper, day-care centers) in assisting their caregiving duties and (3) lived together with their care recipients; these factors predisposed them most towards becoming a ‘24/7’ caregiver, as found in our earlier research [5] (Table 1).

Three key themes were identified:

### 3.1. A World Overturned

The prolonged nature of the pandemic, coupled with the constant changes in restrictions, created a massive upheaval and disruption to PwD’s daily routines. For some, worsening behavioral disturbances such as insomnia led to many caregivers ‘*(losing) sleep at night… for three, four days*’ (CG3). One caregiver describes his mother’s increased care demands that required his attention overnight:


*I want to take a rest a bit, then… [my mother] was in pain, [she] needs me [at night] …needs water… so (I have) no rest the whole day …until the fifth day… my body shuts down.*
(CG3, son)

Caregivers and their care recipients’ lives became intertwined as caregiving responsibilities physically and emotionally dominated the caregiver’s life due to the care recipient’s *‘need for attention 24/7’* (CG3). In particular, CG9’s inability to detach herself from her caregiving responsibilities was exemplified by her incessant around-the-clock worries about the potential harm her mother might face:


*I have to sleep with her. I took a raffia string to tie her hand with mine, so that when she gets up… I will also get up. We’re afraid that the medication makes her so drowsy that when she goes to the toilet, she will slip and fall, or she will jump down.*
(CG9, daughter)

Many caregivers thereby expressed their frustrations that caregiving was often ‘*exhausting*’ and ‘*stressful*’ (CG11), akin to a relentless full-time job that necessitated them to provide round-the-clock care for their loved ones without any respite:


*I always have to be there to watch him take a shower, clean himself, …go to the toilet, …(and) make sure he brushes his teeth properly.*
(CG15, spouse)

### 3.2. Burning on Both Ends

As time and energy are finite resources, the balance between caregiving and one’s personal life eventually became a zero-sum game for many ‘24/7’ caregivers. Caregiving often ‘*ate up a lot of time*’ and it was ‘*difficult to juggle work, personal (commitments and) [caregiving duties] for the elderly*’ (CG4) due to the PwD’s constant need for assistance. Having no choice but to relinquish freedom and control over their pre-existing schedules, many ‘24/7’ caregivers prioritized their loved one’s needs and associated caregiving responsibilities over their personal time and space, highlighting the cultural manifestations of the physical and emotional intensity of caregiving:


*He tends to be so clingy and he’s always in your space. You don’t even have the time to be alone, to have your own thoughts. You have to reschedule your whole life.*
(CG15, spouse)

The negative impact of caregiving became more pronounced when they ignored or deprioritized their personal needs, causing their health, well-being and personal lives to be compromised. Shouldering total responsibility in caregiving with no form of external help, these ‘24/7’ caregivers often ‘*only got (their) own time*’ and space when their care recipients were ‘*resting*’ or ‘*sleep(ing)*’ (CG24). Coupled with the restrictive COVID-19 measures, the emotional support that caregivers usually obtained from their social networks was greatly diminished, with caregivers not being able to ‘*meet up with many people [their friends and family]*’ (CG15) during the pandemic period. Being unable to ‘*sleep*’ or ‘*see (their) friends*’ (CG3) in addition to the mentally, psychologically and physically intensive demands of caregiving, many caregivers stated:


*(It is) difficult to juggle between work, (personal responsibilities) and (caregiving). (So, it can be) consuming in terms of energy and mental health.*
(CG4)

### 3.3. At Wits’ End

‘24/7’ caregivers often resigned themselves to the belief that they exhausted all possibilities to address their problems and ‘*cannot do anything more*’ (CG2) to change their circumstances. Nearing their breaking point, many articulated that they ‘*were going to snap already*’ (CG3), ‘*could not take it*’ (CG9) and were at their wits’ end in handling their loved ones’ difficult behaviors. This subsequently created a sense of lethargy, helplessness and hopelessness at seeking change:


*I am very tired. I am (so) tired that I (can) drop dead.*
(CG11, daughter)

Despite recognizing that ‘*nobody can [provide caregiving] for 24 h*’ (CG11), many remain trapped in seemingly immutable circumstances. Caregivers feel helpless and less in control over their situation, accepting that their caregiving situation will never improve and gradually resigning to fate:


*I have done all sorts of things, suffered all sorts of treatment, I feel so numb. Right now, I can only say, to get through each day one at a time.*
(CG2, spouse)

Unable to see any way out of the perceived insufferable situation due to the care burden associated with providing ‘24/7’ care, the psychological and physical stress proved too much to bear, and one caregiver eventually chose institutionalization as a final resort:


*We had to put her in a nursing home. I lost seven kg within one month. I was very tired and stressed. I couldn’t eat… If I had my choice, I would still want her back, to look after her.*
(CG9, daughter)

## 4. Discussion

This is the first known study on the ‘24/7’ caregiving of PwD and the associated role engulfment in Asia, where ‘24/7’ caregiving is positioned as a nonliteral term encompassing not only caregiving in its physical form but also its emotional and psychological aspects. Indeed, a hallmark of ‘24/7’ caregiving is the time-consuming and pressurizing nature of care responsibilities that restructures and engulfs caregivers’ lives. This is highlighted in our three themes: (1) A World Overturned, (2) Burning on Both Ends and (3) At Wits’ End, which showcase the caregiving stress virtually leaking into the caregiver’s surrounding roles, personal identity and spheres of social life [9]. Our findings fit into the role engulfment model, wherein according to Skaff and Pearlin [23], the role entrenchment and engulfment diminishes the caregiver’s self (loss of self), with relentless role strain, immersion in caregiving and limited social contact wielding a deleterious strain on caregivers’ well-being through negative impacts on mastery and self-esteem, consequently resulting in greater depressive symptomatology [23]. The findings mirror other studies that found that caregivers engulfed in caregiving duties often feel trapped, possess limited mastery and are ill equipped to cope with the incessant demands and emotional upheaval of caregiving as the relentless caregiving demands often overwhelm and stretch the adaptive capacities of the caregiver [5,9,24,25].

One of the factors driving this role engulfment and round-the-clock caregiving is the Asian cultural expectation of familial elder care obligation. This issue is more prevalent and particularly acute in Asian societies (i.e., Singapore) that have the enduring Asian ideologies of familism and filial piety, which denote intense normative feelings of dedication, loyalty, reciprocity, strong solidarity and identification with family members [2,6]. One of the key means to addressing its prevailing eldercare crisis is the embracement of the neoliberal shift to home and community care, where the privatized realm of the home space as the optimal caregiving site [6] and the institutionalization of older parents is strongly stigmatized [26]. In fact, the familial obligation to care for an older loved one is often so culturally ingrained in such Asian societies that caregivers’ struggles often become disregarded and unaddressed [11], with the internalization of such cultural care expectations in caregivers’ daily lives reported to be a stressor contributing to avoidant coping [2], heightened psychological distress and depression in caregivers [10,26,27]. Indeed, these caregivers often find themselves trapped in a ‘caring dilemma’, where emotional and moral forces of reciprocity and familial bonds of affection often motivate their sustained and intense caregiving practices and in which caregivers simultaneously resent the inexorability of their caregiving predicament [11,28].

Such caregiving struggles were made more prevalent in the context of the COVID-19 pandemic. With the nationwide establishment of physical distancing measures and instructions to isolate at home, caregivers’ access to formal and informal caregiver support and respite resources gradually became restricted [13,14,29]. As day-care centers and senior activities centers were closed due to community restrictions, more informal familial care was required. This amplified caregiving responsibilities and frustrations as these caregivers often had to provide full-time care without being able to offload some of the high-intensity caregiving demands onto their regular support systems that were typically available before the pandemic. Inevitably, with round-the-clock caregiving without any respite, many caregivers experienced unprecedented levels of stress, higher depression, anxiety, fatigue and sleep disturbances ‘24/7’ over the past two years [30]. It is therefore essential to support their psychosocial and emotional needs, often beginning with psychoeducation through various healthcare and social service touchpoints to help caregivers realize the unsustainability of their current care provision and its potential detrimental impact on their care recipients’ care. Caregivers can then be guided on expectation management and how to set healthy boundaries to conserve their personal time and space.

However, with caregiving being seen primarily as a familial responsibility, these ‘24/7’ caregivers may hesitate to reach out for support and tap on external sources of help beyond their immediate family nucleus. Indeed, for many Asian family caregivers, caregiving is perceived to be a personal responsibility and feelings of shame and guilt often emerge if one seeks help beyond the family unit, despite the awareness that some caregiving needs are impossible to be fulfilled solely by family members [31,32]. Some caregivers may also have limited knowledge and awareness of community-based services such as day-care, respite care and home-based services, as well as their benefits in reducing caregiver burden, thus leading to the underutilization of social services and caregiver intervention programs [33]. This calls for a whole-of-society initiative that recognizes the importance of sustainability in caregiving and greater community outreach efforts to be integrated into society on a multiscalar level, especially at the neighborhood and grassroots level, to strengthen caregiver support for the greater alleviation of high-intensity caregiving burden. Through caregiver-led support networks, volunteer caregivers can proactively reach out to ‘24/7’ caregivers to break down their barriers and help connect them to a network of informal social support, i.e., a virtual community of like-minded caregiver peers who are going through the same caregiving journey. For instance, WhatsApp and Facebook groups offering caregiver-to-caregiver peer support on caregiving are some of the existing digital platforms in Singapore that older caregivers could utilize to receive and exchange caregiving resources, share their problems and receive social and emotional support. Crucially, this connects them to helpful caregiving resources, information and advice from more experienced caregivers, which according to Friedman and colleagues [34], would help to develop the caregiver’s self-efficacy. This is especially important for ‘24/7’ caregivers who are engulfed by the caregiving role and are not as visible in the community due to their limited usage of formal support services and poor social support networks. Additionally, there is an urgent need for the government and community leaders to enhance the image and heighten public awareness of formal support services to normalize help-seeking behaviors in caregivers, improve the underutilization of social services and uncouple the feelings of guilt and shame stemming from caregivers’ perceived inability to provide the ‘best’ care for their loved ones [35].

## 5. Limitations

As ‘24/7’caregivers make up a subset of the overall caregiver population, our sample remained small as it was part of a wider pool of caregivers who were interviewed. This is because the core characteristics of the ‘24/7’ caregiver were only sieved out through a retrospective analysis of the primary data, with validation from data from an earlier publication [5]. Data saturation was achieved when a continued thematic analysis of the transcripts yielded no new insightful analytical data and information, other than the themes already identified, and when further coding was no longer feasible, resulting in the small sample size. However, our findings are important in pointing towards the presence of this in-need subgroup of caregivers who require much more intensive support from the wider community to enable an effective continued caregiving capacity. A more nuanced reading and in-depth research ought to be conducted on the concept of role engulfment and informal caregiving for PwD on a larger scale to further validate our findings.

## 6. Conclusions

While caregiving can be rewarding and meaningful, it is oftentimes simultaneously stressful, challenging and physically, mentally and socially isolating, especially for ‘24/7’ caregivers who shoulder the caregiving duties and responsibilities alone with little to no avenue for support. This study sheds light on the vulnerabilities associated with ‘24/7’ caregiving; the implications of its physicality and emotionality are gleaned from the lived experiences and struggles faced by ‘24/7’ caregivers as a result of providing continuous care during the COVID-19 pandemic with no recourse to any relief, namely, the upheaval to daily living and routines, the constant prioritization of their loved ones’ needs over theirs which inevitably pushes them towards burnout and the unshakable feelings of helplessness and hopelessness. Cultural factors and norms such as filial obligation to eldercare provision, interwoven and situated in the backdrop of the COVID-19 pandemic with its negative social impacts, influence the mental and physical toll on these caregivers’ well-being. Indeed, there is a crucial need for an urgent and concerted effort to address the caregiver burden and burnout experienced by ‘24/7’ caregivers in accordance with the cultural particularities of an Asian context (Singapore). Increased provisions and accessibility to physical, psychosocial and emotional caregiver support that are integrated at various touchpoints of the healthcare system and into society and the community, especially at the neighborhood and grassroots level, as well as the normalization of help-seeking behaviors in caregivers would enhance their ability to provide sustainable care for their loved ones while feeling empowered and supported by the community around them.

## Figures and Tables

**Table 1 ijerph-19-17046-t001:** Participants’ demographics (N = seven) ^1^.

PID	Age	Sex	Relationship	Living with Care Recipient	Ethnicity	Marital Status	Highest Education Level	Working	Has a Lived-in Domestic Helper
CG2	80	Female	Spouse	Yes	Chinese	Married	Secondary	Retired	No
CG3	62	Male	Son	Yes	Chinese	Widowed	Secondary	Retired	No
CG4	60	Female	Daughter	Yes	Chinese	Single	Secondary	Full time	No
CG9	63	Male	Son in law	Yes	Eurasian	Married	Secondary	Full time	No
CG11	53	Female	Daughter	Yes	Chinese	Single	Secondary	Full time	No
CG15	59	Female	Spouse	Yes	Eurasian	Married	Secondary	Part time	No
CG24	68	Female	Spouse	Yes	Chinese	Married	Degree	Retired	No

^1^ Participants were drawn from a larger qualitative study [16,17].

## Data Availability

The datasets generated and/or analyzed for the current study are available from the corresponding author upon reasonable request.

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
