# Peer review of "‘24/7’ Caregiving: A Qualitative Analysis of an Emerging Phenomenon of Interest in Caregiving"

_ijerph, 2022, doi:10.3390/ijerph192417046_

Round 1

Reviewer 1 Report (Previous Reviewer 2)

‘24/7’ Caregiving: A Qualitative Analysis of An Emerging Phenomenon of Interest In Caregiving

This is a re-submission of an article that I had previously recommended rejection and that authors have re-submitted after implementing important changes to the initial version. I commend their effort to improve the overall quality of the paper, and I believe that the article is now fit for publication. Best wishes.

Reviewer 2 Report (Previous Reviewer 3)

Congratulations on fixes in the article. In my humble opinion, it could be published in the current version.

This manuscript is a resubmission of an earlier submission. The following is a list of the peer review reports and author responses from that submission.

Round 1

Reviewer 1 Report

Thank you for allowing me to review this brief and well-written manuscript.  

Introduction: Despite the relevance of its content, the originality of this manuscript could be improved. In fact, the authors would benefit from reinforcing the specificities of the temporal (pandemic) and cultural (Asian) context of the study, in comparison to existing bodies of knowledge. Indeed, there is a need to adapt best practices in terms of respite, for example, to cultural particularities (asian family values that lead more children to house their parents living with dementia). 

Methods: It would be interesting to 1) specify who performed the analyses and whether co-coding took place; 2) justify the sample size. 

Discussion: The addition of one or two limitations would be appreciated. 

Conclusion: Considering the cultural particularities, the acceptability of assistance outside the family nucleus seems more difficult. Could interventions, such as a virtual community of practice of caregivers to promote support between peers, be a more acceptable intervention? 

Reviewer 2 Report

Thank you for the opportunity to review this article. This is an interesting topic, but the article has several major flaws that impede its publication in its current form. I personally believe that it doesn’t have sufficient quality to be published: sample size is very small even for a qualitative study, results don’t follow any of the consolidated criteria for reporting qualitative research (COREQ), they are presented very briefly and are not scientifically sound. The reference list is very short, introduction and discussion sections are completely insipient. Considering all criteria above, I believe this is not a suitable article for the journals readers.

Reviewer 3 Report

In overall, I have to say the manuscript is well-written and seems to fit perfectly to the theme of the special issue. However, a few minor, but important improvements have to be made before the eventual publication.

Title:

the title should precisely describe the type of the research that You have performed.

Materials and methods:

 Further details are in our previous 59 publication [5].” What do You think about extending the current version of material and methods, as it seems to be relatively sparse? Maybe more broad explanation would be of benefit for some potential further readers?

Results:

You should better describe Table 1. Wat does “Form of Help “ means and why all participants got “none”? What “working” means? How someone can work full-time and be a caregiver really for 24/7?

Discussion: Discussion is also relatively short, I would extend it. I would add some pieces in relation to the conclusions, however, first proper conclusions should be drawn

Conclusions:

In overall information from the conclusions seems to be interesting. However, I would add references to all of those statements (or delete them if there are none) and move them to the discussion. In my humble opinion, in conclusions You should stick mainly to the conclusions drawn from Your own study.

Abstract:

After all of the changes please fix the abstract accordingly.
